# Anti-inflammatory effects of eupatilin on *Helicobacter pylori* CagA-induced gastric inflammation

**Bong Eun Lee**[1,2‡], **Su Jin Park**[2‡], **Gwang Ha Kim**[1,2]*, **Dong Chan Joo**[1,2], **Moon Won Lee**[1,2]

1 Department of Internal Medicine, Pusan National University School of Medicine, Busan, Korea,
2 Biomedical Research Institute, Pusan National University Hospital, Busan, Korea

‡ BEL and SJP contributed equally to this work as co-first authors on this work.
* doc0224@pusan.ac.kr

**Data Availability Statement:** All relevant data are within the manuscript and supporting information files.

**Funding:** This study was supported by the Dong-A Pharmaceutical Co. Ltd., Seoul, Korea, and a

## Abstract

### Background

Eupatilin, a flavone isolated from *Artemisia* species, exerts anti-inflammatory, anti-oxidative, and anti-neoplastic activities. However, the effects of eupatilin on *H. pylori*-associated gastritis remain unclear. Thus, this study aimed to investigate the anti-inflammatory effects of eupatilin on gastric epithelial cells infected with cytotoxin-associated gene A (CagA)-positive *Helicobacter pylori*.

### Materials and methods

AGS human gastric carcinoma cells were infected with a CagA-positive *H. pylori* strain and then treated with 10, 50, or 100 ng of eupatilin. After 24 h, the expression levels of CagA, phosphoinositide 3-kinase 1 (PI3K), nuclear factor (NF)-κB, interleukin (IL)-1β, and tumor necrosis factor (TNF)-α in the cell lysates were measured using western blotting, and the mRNA levels of IL-6, IL-8, and monocyte chemoattractant protein (MCP)-1 were measured using real-time polymerase chain reaction.

### Results

CagA translocation into AGS cells resulted in an elongated cell morphology, which was significantly suppressed by eupatilin treatment in a dose-dependent manner. Immunofluorescence staining for anti-CagA showed that eupatilin treatment dose-dependently inhibited CagA expression in the *H. pylori*-infected AGS cells. *H. pylori* infection increased the levels of pro-inflammatory cytokines including IL-1β, TNF-α, IL-6, IL-8, and MCP-1, and eupatilin treatment significantly reduced the levels of these cytokines in a dose-dependent manner. Additionally, eupatilin treatment dose-dependently suppressed the expression of PI3K and NF-κB.

National Research Foundation of Korea (NRF) grant funded by the Korean government (MSIT) (No. NRF- 2022R1A5A2027161). The funders had no role in study design, data collection and analysis, decision to publish, or preparation of the manuscript.

**Competing interests:** The authors have declared that no competing interests exist.

## Conclusions

Eupatilin treatment demonstrated anti-inflammatory effects on CagA-positive *H. pylori*-infected gastric epithelial cells by inhibiting CagA translocation, thereby suppressing the NF-κB signaling pathway. These results suggest that eupatilin plays a protective role against CagA-positive *H. pylori*-induced gastritis.

## Introduction

*Helicobacter pylori (H. pylori)* causes chronic inflammation of the gastric mucosa (gastritis), which can progress to major gastric diseases, including peptic ulcer, gastric cancer, and gastric mucosa-associated lymphoid tissue lymphoma [1]._ENREF_1 Cytotoxin-associated gene A (CagA) is an important virulence factor of *H. pylori*, and CagA translocation into gastric epithelial cells induces cellular transformation to an elongated shape, referred to as the hummingbird phenotype, which is characterized by one or more protrusions on the cell membrane; this phenotype causes a wide array of alterations in cellular signaling, which leads to the infiltration of neutrophils and lymphocytes as well as the increment of pro-inflammatory cytokines [2]. CagA activates the nuclear factor (NF)-κB signaling pathway, which promotes the expression of pro-inflammatory cytokines, such as interleukin (IL)-1β, IL-6, IL-8, and tumor necrosis factor (TNF)-α, ultimately causing persistent gastric inflammation [3]. *H. pylori* CagA-induced chronic inflammatory response is a strong risk factor for developing peptic ulcer and gastric cancer; therefore, inhibiting these inflammatory cascades is essential for managing *H. pylori*-associated gastric diseases.

Eupatilin (5,7-dihydroxy-3,4,6-trimethoxyflavone, available as a commercial drug, Stillen$^{\textregistered}$), a flavone isolated from *Artemisia* species, exerts anti-inflammatory, anti-oxidant, and anti-cancer activities [4, 5]. Flavonoids, a class of polyphenolic compounds in various plants, display bioactive properties in human cell lines [6]. *Artemisia* leaves have a rich history of traditional use in addressing inflammatory and ulcerogenic disorders in Korea [7]. Eupatilin is a pharmacologically active compound isolated from *Artemisia asiatica* extract [8]. It reduces the levels of pro-inflammatory cytokines TNF-α, IL-6, and IL-1ß by suppressing the NF-κB-mediated signaling pathway [9, 10]. *In vitro* studies reported that eupatilin protects gastric epithelial cells from indomethacin-induced oxidative cellular damage [11] and ethanol-induced gastric mucosal injury by inhibiting inflammation, enhancing gastric mucosal defense, and ameliorating oxidative stress [12].

Despite substantial evidence regarding the anti-inflammatory effects of eupatilin, little is known about the effects of this compound on *H. pylori*-associated gastric inflammation. Therefore, we aimed to investigate the effects of eupatilin on the production of pro-inflammatory cytokines and signaling pathways in gastric epithelial cells infected with CagA-positive *H. pylori*.

## Materials on methods

### *H. pylori* strain and culture

*H. pylori* ATCC 43504 strain (CagA-positive and vacA s1-m1 type strain) was obtained from the American Type Culture Collection (ATCC; Rockville, MD, USA). *H. pylori* was cultured under microaerophilic conditions (5% $O_2$, 10% $CO_2$, and 85% $N_2$) on a chocolate agar plate (Synergy Innovation, Korea) at 37°C for 3 days. After incubation for 3 days, 200 μL of brain

heart infusion supplemented with 5% fetal bovine serum (FBS; Gibco, Grand Island, NY, USA) was added to the desired amount of chocolate agar in a 1.5 ml Eppendorf tube.

## Cell culture and *H. pylori* infection

AGS human gastric carcinoma cells (ATCC CRL-1739) were purchased from the ATCC (Rockville). AGS cells were seeded in Roswell Park Memorial Institute 1640 medium (Gibco, Grand Island, NY, USA) supplemented with 10% inactivated FBS and 1% penicillin (Sigma, Burlington, MA, USA), and then cultured in a humidified atmosphere containing 5% $CO_2$ at 37˚C. Upon reaching 70% confluence, the cells were incubated in serum-free medium for 24 h and then infected with *H. pylori* at a multiplicity of infection of 200:1.

## Drug treatment

Eupatilin supplied by Dong-A Pharmaceutical Co. Ltd. (Seoul, Korea) was dissolved in 10% dimethyl sulfoxide. AGS cells were treated with 10, 50, and 100 ng of eupatilin within 24 h of *H. pylori* infection.

## Cell viability (cytotoxicity assay)

AGS cells were seeded at $1 \times 10^3$ cells/well in 96-well plates and then treated with 10, 50, and 100 ng of eupatilin for 24 h. Cell viability was measured using a Cell Counting Kit-8 assay (CCK-8; Enzo Life Science, NY, USA) in accordance with the manufacturer's instructions. The plate was added with 10 μL of the CCK-8 reagent and then incubated at 37˚C for 2 h. The absorbance at 450 nm was measured using a microplate reader (JSBIO, Seoul, Korea). Cell viability was expressed as the relative absorbance of the treated AGS cells to the untreated cells (control).

## Morphological changes after *H. pylori* infection

Morphological changes in AGS cells were examined after 24 h of *H. pylori* infection and after eupatilin treatment. Cells showing an elongated hummingbird phenotype were counted in five fields in three dishes.

## Immunofluorescence staining

Immunofluorescence staining was performed to identify CagA-positive AGS cells after *H. pylori* infection. Briefly, AGS cells were seeded at $1 \times 10^4$ cells/well onto a 12 mm coverslip glass at the bottom of 24-well transwell plates. After *H. pylori* stimulation, the cells were treated with 10, 50, and 100 ng of eupatilin for 24 h. The cells were washed with phosphate-buffered saline (PBS), fixed with 4% formaldehyde for 20 min at 4˚C, and then permeabilized with 0.1% Triton X-100 for 15 min at room temperature. Subsequently, they were blocked for 30 min in a blocking solution and then incubated for 1.5 h with the primary antibody against CagA (Santacruz, Dallas, USA). Finally, the cells were washed with PBS, incubated with Alexa Fluor 594-conjugated anti-mouse secondary antibody (Invitrogen, Carlsbad, CA, USA), and then stained with 0.5 μg/ml DAPI solution (Abcam, Cambridge, UK) for nuclear staining. Cell images were taken in 4–6 fields in three wells and evaluated using an Eclipse Ti-E fluorescence microscope (Nikon, Tokyo, Japan) at 20× magnification.

## Western blot

After being treated with eupatilin for 24 h, AGS cells were harvested and lysed in lysis buffer [pH 7.6, 4 mM 4-(2-aminoethyl) benzenesulfonyl fluoride hydrochloride, 2 mM benzamidine,

10 µM leupeptin, 10 µM pepstatin A, 1 mM EDTA, 10 µM EGTA, and phosphate inhibitors (Translab, Daejeon, Korea)]. The cells were analyzed using a BCA protein assay kit (Thermo Fisher Scientific, Waltham, MA, USA). Samples containing the same amount of protein (50 µg) were separated by 4%–15% sodium dodecyl sulfate-polyacrylamide gel electrophoresis and transferred onto a polyvinylidene fluoride (PVDF) membrane. The PVDF membrane was blocked with 1% bovine serum albumin solution at 15°C for 1.5 h and then incubated overnight at 4°C with one of the following primary antibodies: Cag A (1:1000; Santacruz), phosphoinositide 3-kinase (PI3K; 1:1000; Santacruz), NF-κB (1:1000; Cell Signaling, Danvers, USA), TNF-α (1:1000; Abcam, Cambridge, UK), IL-1ß (1:1,000; Cell Signaling), and glyceraldehyde-3-phosphate dehydrogenase (GAPDH; 1:1000; Santacruz) as an internal control. The following day, the membrane was placed in tris buffered saline-tween20 solution, washed at 15°C for 1.5 h, and then incubated at 15°C for 2 h with a secondary antibody (mouse anti-rabbit IgG-HRP, anti-mouse IgG-HRP, Santa Cruz). Finally, an enhanced chemiluminescence western blotting kit (SmartGene ECL High Femto Solution; SmartGene Soft, Daejeon, Korea) was used to detect the markers. The optical density of the protein bands was determined using ImageJ 1.52 software. All proteins were quantified relative to GAPDH, and the quantified values were presented in a bar graph.

## Analysis of mRNA expression

Total cellular RNA was extracted from the eupatilin-treated AGS cells through reverse transcription polymerase chain reaction (RT-PCR) using a RNeasy plus mini kit (QIAGEN, Hilden, Germany) in accordance with the manufacturer's protocols. Briefly, 2 µg of total RNA was mixed with 1 µg of oligo dT (Promega, Wisconsin, USA) in a total volume of <15 µL, and the mixture was heated at 70°C for 5 min. Subsequently, 1.25 µl of dNTP (Promega) and M-MLV reverse transcriptase (Promega) were added to the heated mixture in a total volume of 20 µL. cDNA was synthesized by reverse transcription at 42°C for 60 min and at 95°C for 5 min. The PCR primer sequences used were as follows IL-6 (5′–TCGTGGAAATGAGAAAAGAG TTG–3′; 5′–GACCACAGTGAGGAATGTCCAC–3′) [13], IL-8 (5′–AGGGTTGCCAGATGCAA TAC–3′; 5′–AAACCA AGGCACAGTGGAAC–3′) [14], and MCP-1 (5′–CCAAAGAAGCTGT AGTTTTTGTC–3′; 5′–GCATTAGCTTCAGATTTACGG–3′) [13]. GAPDH (5′–CACCTTCTG CAAAATTATGGCG-3′; 5′–ACCTTTGCCAAGTCTAACTGTTAA–3′) [15] was used as the internal control. Real-time PCR was performed using 12.5 µL of 2X TOPsimple™ DyeMIX--Tenuto (Enzynomics, Korea) and 2 µL of cDNA template in a final volume of 25 µL. The mixture was incubated at 94°C for 4 min, followed by 32 cycles of PCR amplification. The PCR program was as follows: denaturation at 95°C for 30 s; annealing at a transitional temperature range of 54–59°C, with an increase of 0.5°C per cycle; and an extension at 72°C for 30 s. A final extension was performed at 72°C for 30 s, followed by an additional 7 min at 72°C after each cycle. After the final cycle, melting-point analysis of the samples was performed at 54–94°C. The optical density of the RNA bands was determined using ImageJ 1.52 software. All mRNAs were quantified relative to GAPDH, and the quantified values were presented in a bar graph.

## Statistical analysis

Data are expressed as the mean ± standard deviation, and statistical analyses were performed using Student's $t$-test, with a significance threshold set at a probability level of 0.05.

## Results

### Effects of eupatilin on CagA translocation into AGS cells and CagA-induced morphologic changes

The effect of eupatilin on cell viability was evaluated first. Eupatilin treatment results in a survival rate of >90% in the CCK-8 assay and did not show significant cytotoxicity even at a high concentration of 100 ng (Fig 1). CagA translocation into the AGS cells induced the elongated hummingbird phenotype (Fig 2A), which is a distinctive morphologic changes induced by CagA through several references [16, 17]. Eupatilin treatment significantly suppressed the production of hummingbird cells in a dose-dependent manner (Fig 2A and 2B). Immunofluorescence staining for anti-CagA also showed that eupatilin treatment dose-dependently inhibited CagA expression in the *H. pylori*-infected AGS cells (Fig 3).

### Anti-inflammatory effects of eupatilin on *H. pylori* CagA-infected AGS cells

Western blot results on the protein expression of IL-1β and TNF-α in the *H. pylori*-infected AGS cells before and after eupatilin treatment are shown in Fig 4A. RT-PCR results on the mRNA levels of IL-6, IL-8, and MCP-1 in the *H. pylori*-infected AGS cells before and after eupatilin treatment are shown in Fig 4B. *H. pylori* infection increased the levels of pro-inflammatory cytokines IL-1β, TNF-α, IL-6, IL-8, and MCP-1, and eupatilin treatment significantly

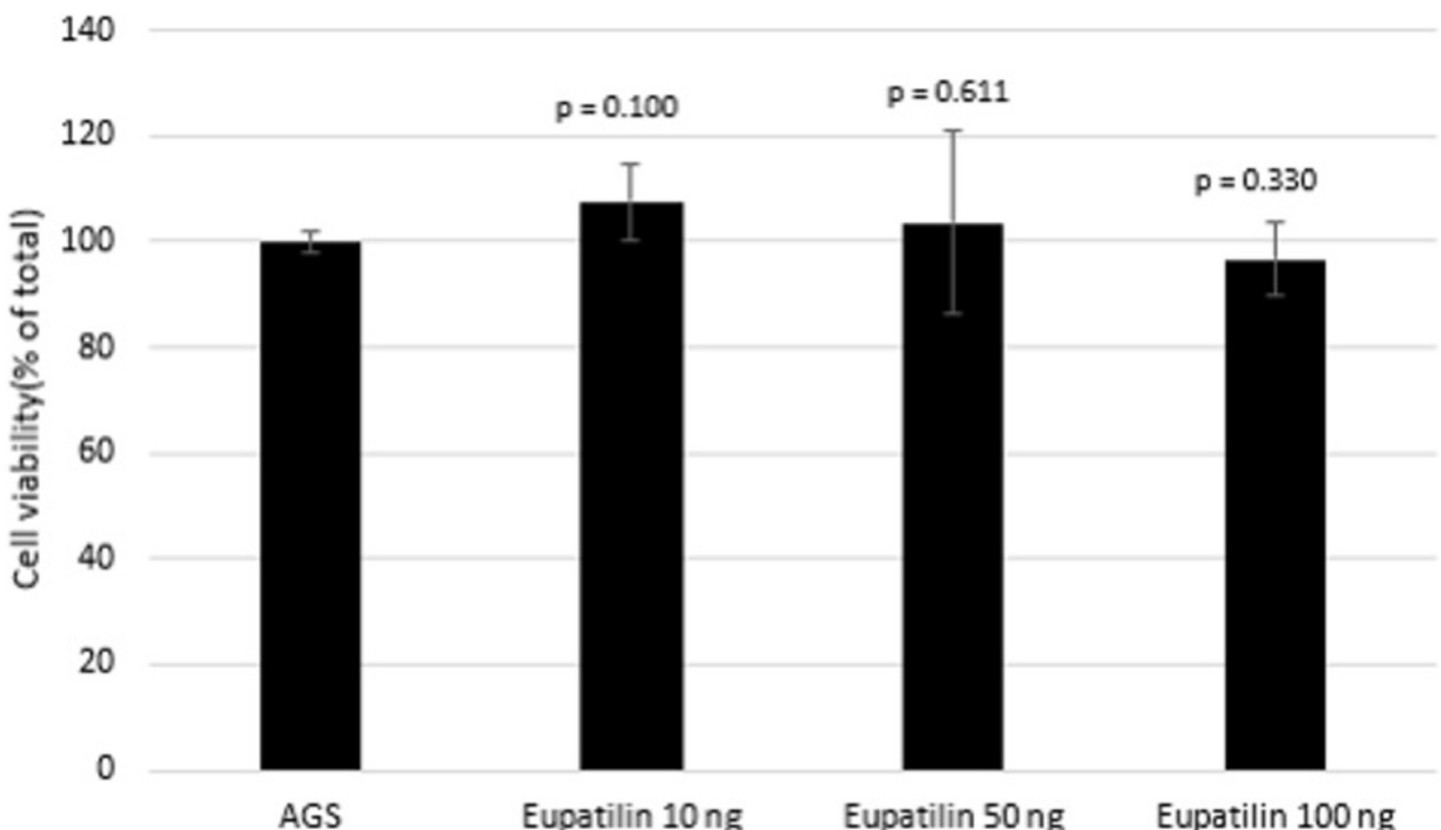

**Fig 1. Cytotoxicity of eupatilin on human AGS gastric carcinoma cells.** AGS cells were treated with eupatilin (10, 50, or 100 ng) for 24 h, and their viability was determined using a Cell Counting Kit-8 assay (Enzo Life Science, NY, USA). HP = *Helicobacter pylori*.

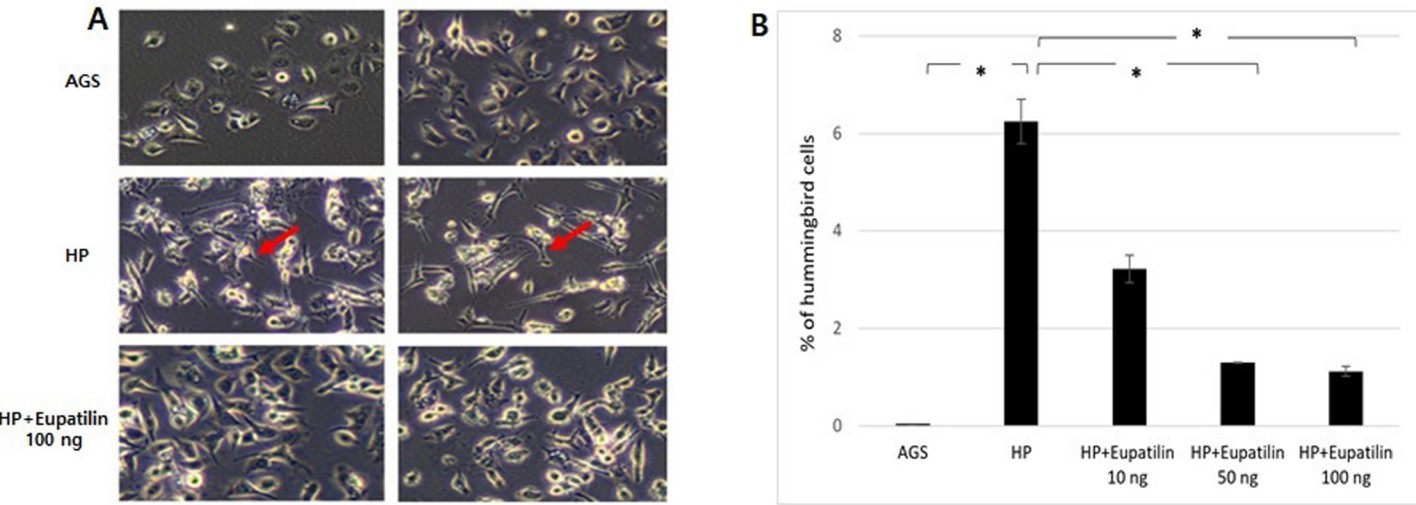

**Fig 2. Morphologic changes (hummingbird cells) induced by *Helicobacter pylori* CagA translocation.** (A) CagA translocation into AGS cells increased the number of hummingbird cells (red arrows), and treatment with eupatilin (100 ng) inhibited the production of hummingbird cells. (B) Eupatilin reduced the number of hummingbird cells in a dose-dependent manner. HP = *Helicobacter pylori*.

reduced the levels of these inflammatory markers in a dose-dependent manner (Fig 4C–4G). These findings suggest that eupatilin exhibits anti-inflammatory effects by effectively suppressing the expression and production of pro-inflammatory cytokines in *H. pylori*-infected gastric epithelial cells.

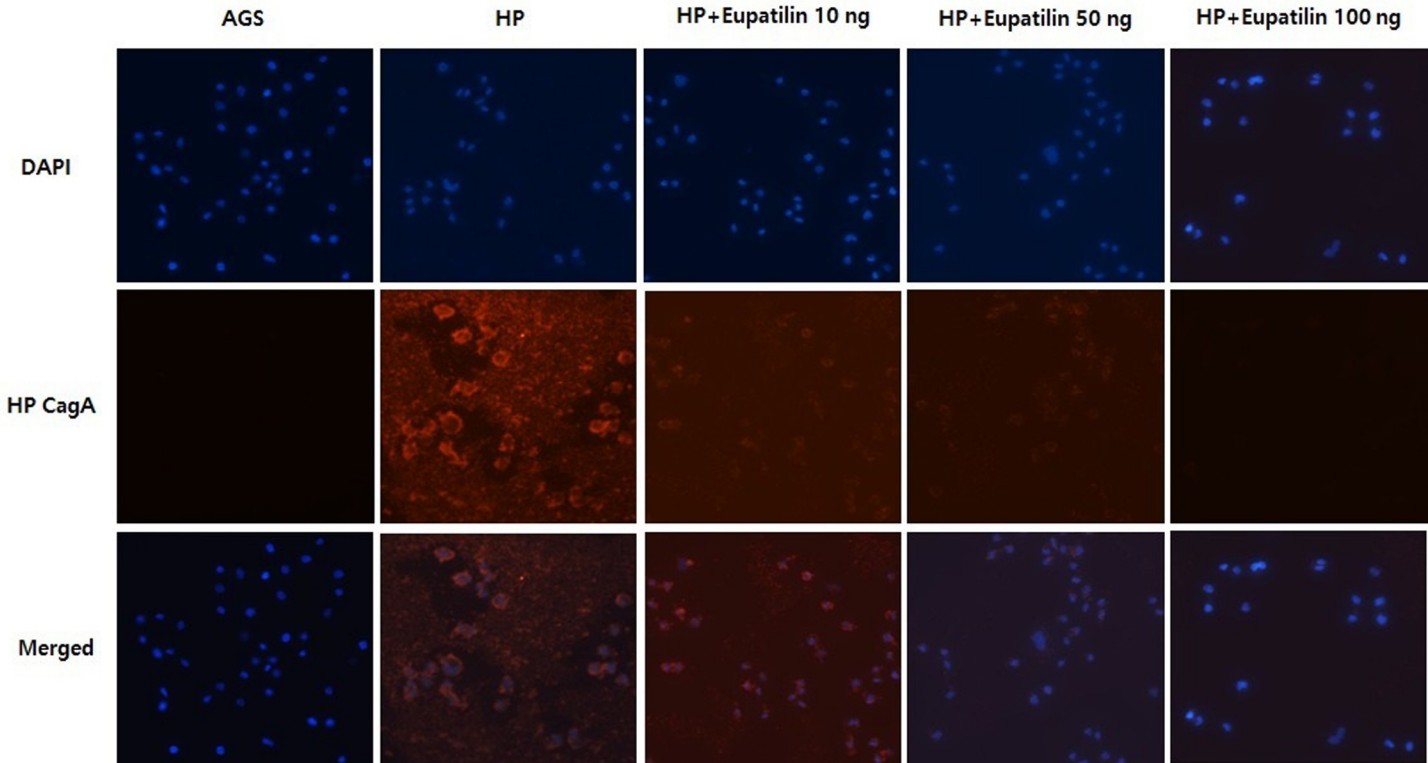

**Fig 3. Immunofluorescence staining with anti-CagA antibody and DNA counterstaining with DAPI (blue).** Eupatilin showed dose-dependent inhibitory effects on CagA expression in *Helicobacter pylori*-infected AGS cells (magnification, 20×). HP = *Helicobacter pylori*.

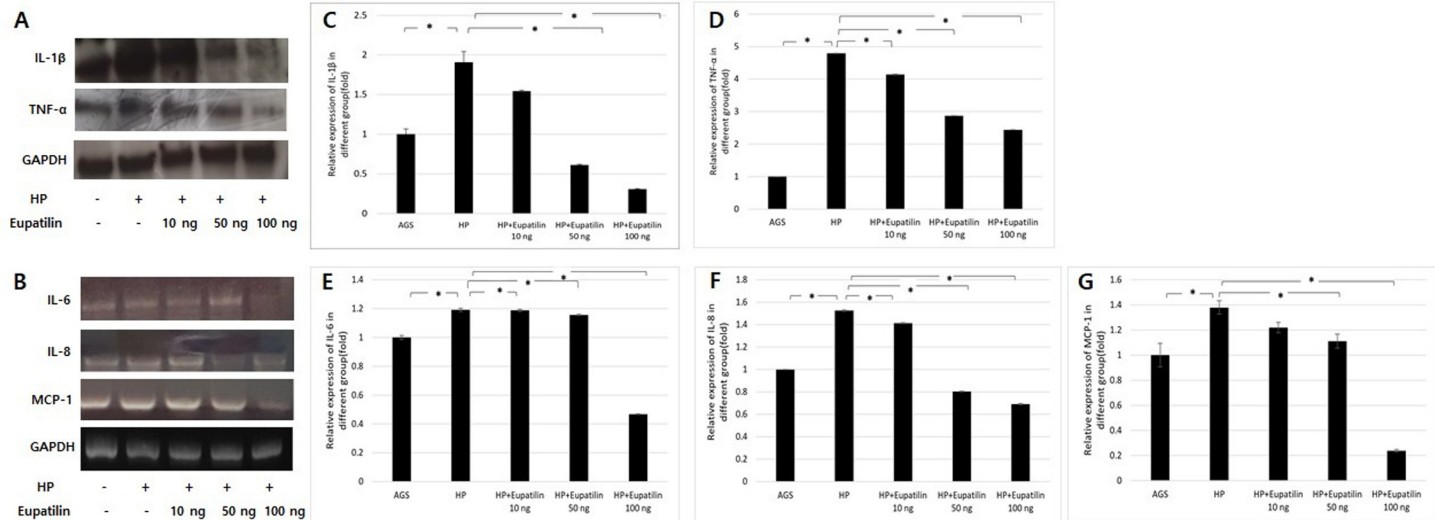

**Fig 4. Eupatilin inhibited pro-inflammatory cytokines in *Helicobacter pylori*-infected AGS cells.** The protein levels of IL-1β and TNF-α were determined by western blot (A), and the mRNA expression of IL-6, IL-8, and MCP-1 was determined using RT-PCR (B). GAPDH was used as an internal control. Statistical significance of IL-1 β (C), TNF-α (D), IL-6 (E), IL-8 (F), and MCP-1 (G) was analyzed using Student's *t*-test. * $p < 0.05$. HP = *Helicobacter pylori*.

## Anti-inflammatory effects of eupatilin through inhibiting *H. pylori* CagA-induced NF-κB signaling pathway in AGS cells

CagA was not expressed in the AGS cells without *H. pylori* infection. CagA expression increased in the cells with *H. pylori* infection, but this expression dose-dependently decreased in the infected cells after eupatilin treatment (Fig 5A and 5B). Additionally, eupatilin treatment dose-dependently suppressed the expression of PI3K and NF-κB (Fig 5C and 5D). These results suggest that eupatilin inhibits *H. pylori*-induced inflammatory responses by inhibiting the CagA/PI3K/NF-kB signaling pathway.

## Discussion

*H. pylori* infection is an important cause of chronic gastritis, which can progress to peptic ulcer disease or gastric malignancies, including adenocarcinoma and mucosa-associated lymphoid tissue lymphoma [18]. In particular, *H. pylori*-induced chronic gastritis is the first step in the multistep cascade of gastric adenocarcinoma [19]. CagA is the most important virulence factor of *H. pylori* because it determines the severity of gastric inflammation and the outcomes of infection. CagA translocation from bacteria into gastric epithelial cells through a type IV secretion system activates NF-κB, a major pro-inflammatory signaling pathway that promotes the release of pro-inflammatory cytokines involved in proliferation, angiogenesis, invasion, and blockade of apoptosis [20]. CagA also triggers inflammation and oxidative stress via various pathways, including IL-11/signal transducer and activator of transcription (STAT)-3/ CDX2, c-Myc/p21/extracellular signal regulated kinase (ERK)-mitogen-activated protein kinase (MAPK), toll-like receptor, and reactive oxygen species/apoptosis signal regulating kinase-1/c-Jun N-terminal kinase signaling [21], which leads to DNA damage and genomic instability, ultimately increasing the risk of gastric cancer development. Vacuolating cytotoxin A, urease, flagellum, catalase, and superoxidase dismutase are other *H. pylori* virulence factors that facilitate carcinogenesis and enable colonization and proliferation. These factors not only induce inflammatory responses but also control and regulate these responses, maintaining chronic inflammation and ultimately inducing malignant alteration [22]. Antibiotic therapies

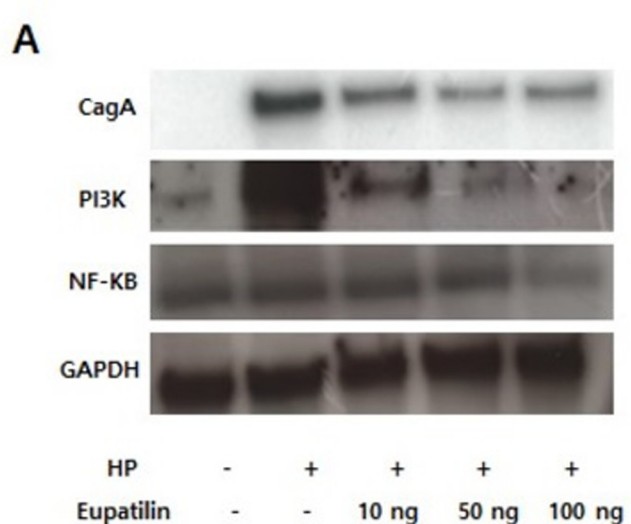

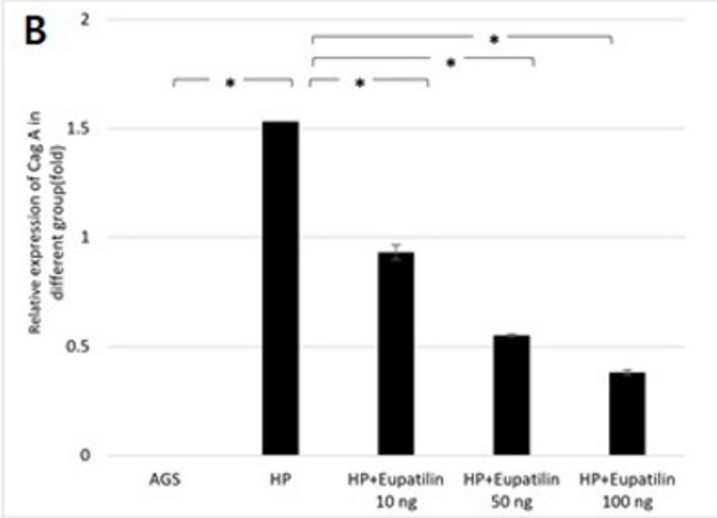

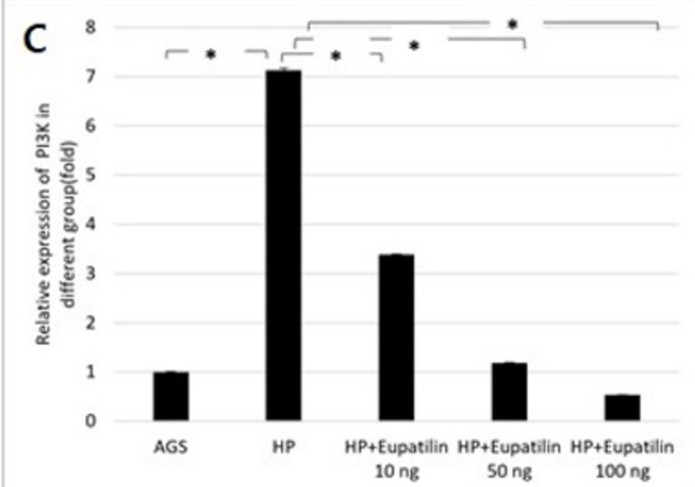

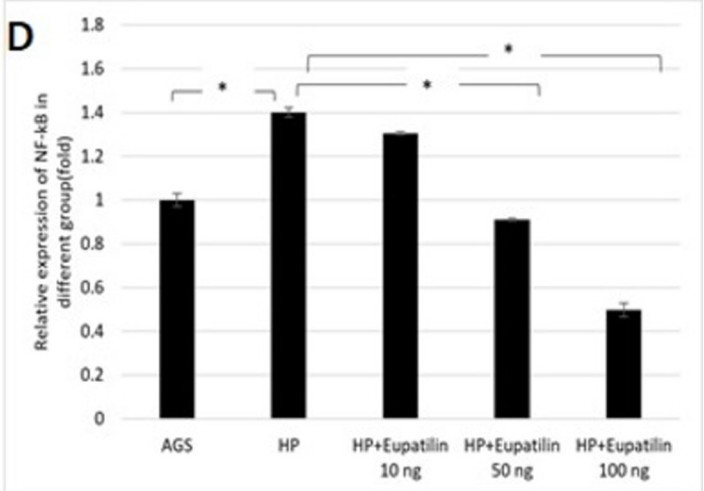

**Fig 5. Eupatilin exerted anti-inflammatory effects by inhibiting the CagA/PI3K/NF-kB signaling pathway in *Helicobacter pylori*-infected AGS cells.** The protein expression of CagA, PI3K, and NF-kB was determined using western blot (A). GADPH was used as an internal control. Statistical significances of CagA (B), PI3K (C), and NF-kB (D) were analyzed using Student's *t*-test. $^*$ $p < 0.05$. HP = *Helicobacter pylori*.

are the primary treatment for *H. pylori* infection. However, persistent increase in antibiotic resistance worldwide and a possible gastric or gut dysbiosis after *H. pylori* eradication complicate the treatment of *H. pylori* infection. Furthermore, some individuals are intolerant to antibiotics, and chronic inflammation and long-term tissue damage may persist even after *H. pylori* eradication. These findings highlight the need for protective agents against *H. pylori*-induced gastritis. In the present study, eupatilin exhibited anti-inflammatory properties in CagA-positive *H. pylori*-infected gastric epithelial cells. This study suggests a safe and effective alternative therapeutic approach that employs a natural compound to treat *H. pylori*-induced gastritis.

Eupatilin prevents tissue damage through its anti-inflammatory, anti-oxidant, anti-cancer, and anti-microbial activities. It also reduces TNF-α-induced IL-8 and chemokine (C-C motif) ligand 20 (CCL20) production by blocking the p38 kinase and NF-κB pathways in human

gastric epithelial AGS cells, suggesting that the gastroprotective effect of eupatilin might be linked to the NF-κB signaling pathway [23]. In a rat animal model, eupatilin effectively ameliorates ethanol-induced gastro-hemorrhagic lesions by suppressing gastric lipid peroxidation and xanthine oxidase activity up to normal levels, implying that the major anti-gastritis mechanism of eupatilin is radical scavenging [24]. In addition, high-dose eupatilin treatment improves chronic erosive gastritis in Sprague–Dawley rats by decreasing erosion length, reducing taurocholate-induced intestinal fibrosis, and increasing glutathione content in a dose-dependent manner [25, 26]. Subsequent clinical studies showed that eupatilin has better efficacy and safety than misoprostol for the treatment of nonsteroidal anti-inflammatory drug-associated gastroduodenal injury [27, 28]. Another clinical trial compared the efficacy and safety of eupatilin and cetraxate in 512 patients with erosive gastritis and obtained significantly higher cure rates without significant serious adverse events in eupatilin-treated patients than in cetraxate-treated patients [29].

Despite substantial evidence regarding the anti-inflammatory effects of eupatilin, little is known about the effects of this compound on *H. pylori*-associated gastritis. Ko et al. reported that eupatilin derivative 7-carboxymethyloxy-3′,4′,5-trimethoxy flavone exerts anti-inflammatory activity in *H. pylori*-infected gastric epithelial cells by inducing the dissociation of the IKK-γ–Hsp90 complex and suppression of NF-κB signaling [30]. *H. pylori* CagA promotes NF-κB through multiple signaling pathways via activating Ras/Raf, P21-activated kinase 1, PI3K, or TNF-α-associated factor 6/transforming growth factor-β-activated kinase 1 [31]. Moreover, eupatilin can inhibit the NF-κB-mediated inflammatory response and suppress cytokine production, including IL-6 and IL-1β, in response to lipopolysaccharide [23]. In the present study, we determined the inhibitory effect of eupatilin on *H. pylori* CagA-induced gastritis, focusing on the CagA/PI3K/NF-κB signaling pathway. CagA stimulates PI3K signaling that affects Akt, and then Akt activates NF-κB through the activation of IκB kinase, finally promoting the transcription of pro-inflammatory cytokines. We initially identified morphological changes associated with CagA-induced inflammation in gastric epithelial cells. Results showed that eupatilin treatment reversed the *H. pylori* CagA-induced cellular elongation (hummingbird cells) by alleviating CagA translocation from *H. pylori* into AGS cells. In addition, eupatilin dose-dependently suppressed the expression of pro-inflammatory cytokines, such as TNF-α, IL-1β, IL-6, IL-8, and MCP-1, by downregulating the PI3K/NF-κB-mediated inflammatory pathway. These findings suggest that eupatilin effectively mitigates *H. pylori* CagA-induced gastritis by inhibiting the production of pro-inflammatory cytokines and suppressing CagA/PI3K/NF-κB signaling.

This study has some limitations. We employed AGS human gastric carcinoma cell line instead of using normal gastric cell line, though AGS cells already could possess a heightened level of inflammation before *H. pylori* infection. Nevertheless, AGS cells have strong cell viability and have been extensively employed to understand the consequences of infection on epithelial cells, particularly in experiments on the response to *H. pylori* [16]. Therefore, we thought that AGS cells can serve as an alternative model for normal gastric epithelial cell in this research. Next, despite convincing *in vitro* evidence of the anti-inflammatory effects of eupatilin on *H. pylori* CagA-infected gastric epithelial cells, the exact mechanisms of action of eupatilin *in vivo* remain unclear. The interaction of *H. pylori* with the host involves complex host–environment factors that could not be fully replicated in *in vitro* studies. Thus, further *in vivo* studies are needed to determine the safe and effective dose of eupatilin to exert anti-inflammatory effects against *H. pylori*-induced gastritis. Moreover, we did not confirm the direct effect of eupatilin on PI3K/NF-κB signaling. We initially focused on the inhibitory effect of eupatilin on *H. pylori* CagA, a factor known to activate the PI3K/NF-κB signaling pathway, rather than on its direct activity on PI3K/NF-κB pathways. We determined that eupatilin

treatment dose-dependently inhibited CagA expression, which could sufficiently suppress CagA-induced PI3K/ NF-κB pathways. Furthermore, in addition to the NF-κB signaling pathway, other signaling pathways, including ERK/MAPK and Janus kinase/STAT, are involved in gastritis and carcinogenesis induced by *H. pylori* infection. Future research should be conducted to reveal other anti-inflammatory pathways of eupatilin and identify whether it exerts a direct anti-tumor activity in preventing or treating gastric cancer. Finally, AGS, a gastric epithelial cell line derived from a patient with gastric adenocarcinoma, was used in this study. This is an important consideration because the metabolome of AGS cells, which are transformed cells isolated from cancer cells, may not accurately reflect the metabolome of normal gastric epithelial cells. Nevertheless, *H. pylori*-infected AGS cells can serve as a model for normal gastric epithelial cells exposed to *H. pylori*, given that AGS cells have been extensively employed in experiments on the response to *H. pylori*.

In conclusion, eupatilin exerts anti-inflammatory activities in CagA-positive *H. pylori*-infected gastric epithelial cells by inhibiting CagA translocation, thereby suppressing the NF-κB signaling pathway. These results suggest that eupatilin plays a protective role against CagA-positive *H. pylori*-induced gastritis.

## Supporting information

**S1 Fig. Raw images of GAPDH in Fig 4A.**
(TIF)

**S2 Fig. Raw images of IL-1 β in Fig 4A.**
(TIF)

**S3 Fig. Raw images of TNF-α in Fig 4A.**
(TIF)

**S4 Fig. Raw images of GAPDH in Fig 4B.**
(TIF)

**S5 Fig. Raw images of IL-6 in Fig 4B.**
(TIF)

**S6 Fig. Raw images of IL-8 in Fig 4B.**
(TIF)

**S7 Fig. Raw images of MCP-1 in Fig 4B.**
(TIF)

**S8 Fig. Raw images of CagA in Fig 5A.**
(TIF)

**S9 Fig. Raw images of NF-kB in Fig 5A.**
(TIF)

**S10 Fig. Raw images of PI3K in Fig 5A.**
(TIF)

**S11 Fig. Western blot images with weight markers in Figs 4A and 5A.**
(PPTX)

**S1 File. Mean, standard deviation, and p-values to build graphs in figures.**
(XLSX)

## Author Contributions

**Conceptualization:** Gwang Ha Kim.

**Data curation:** Bong Eun Lee, Su Jin Park, Dong Chan Joo.

**Formal analysis:** Bong Eun Lee, Su Jin Park, Moon Won Lee.

**Funding acquisition:** Gwang Ha Kim.

**Writing – original draft:** Bong Eun Lee, Su Jin Park.

**Writing – review & editing:** Bong Eun Lee, Su Jin Park, Gwang Ha Kim, Dong Chan Joo, Moon Won Lee.

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
