## [Decision Letter · Decision Letter 0]

17 Jul 2024

PONE-D-24-23616

Anti-inflammatory effects of eupatilin on Helicobacter pylori CagA-induced gastric inflammation

PLOS ONE

Dear Dr. Kim,

Thank you for submitting your manuscript to PLOS ONE. After careful consideration, we feel that it has merit but does not fully meet PLOS ONE’s publication criteria as it currently stands. Therefore, we invite you to submit a revised version of the manuscript that addresses the points raised during the review process.

We look forward to receiving your revised manuscript.

Kind regards,

Lucia Magnelli

Academic Editor

PLOS ONE

Journal Requirements:

 "This study was supported by the Dong-A Pharmaceutical Co. Ltd., Seoul, Korea, and a National Research Foundation of Korea (NRF) grant funded by the Korean government (MSIT) (No. NRF- 2022R1A5A2027161)."

Reviewers' comments:

Reviewer's Responses to Questions

**Comments to the Author**

1. Is the manuscript technically sound, and do the data support the conclusions?

Reviewer #1: Partly

Reviewer #2: Partly

Reviewer #3: Yes

2. Has the statistical analysis been performed appropriately and rigorously? 

Reviewer #1: I Don't Know

Reviewer #2: Yes

Reviewer #3: Yes

3. Have the authors made all data underlying the findings in their manuscript fully available?

Reviewer #1: Yes

Reviewer #2: Yes

Reviewer #3: Yes

4. Is the manuscript presented in an intelligible fashion and written in standard English?

Reviewer #1: Yes

Reviewer #2: Yes

Reviewer #3: Yes

5. Review Comments to the Author

Reviewer #1: Authors of the manuscript: Anti-inflammatory effects of eupatilin on Helicobacter pylori CagA-induced gastric inflammation. They present the results of studies on the effect of eupatilins on the infectivity of CagA-positive H. pylori against AGS human gastric carcinoma cells. The research included the effect of eupatilins on the induction pathway of pro-inflammatory factors induced by the presence of the CagA toxin.

1.The introduction and discussion need to be completely revised. The authors poorly outlined the background in the introduction, many elements that should have been included in the introduction are included in the discussion.

2.Materials and methods instead of materials on methods

3.Please correct the research objective.

4.The results should be better discussed.

5.In the chapter: Anti-inflammatory effects of eupatilin through inhibition of H. pylori CagA-induced NF-κB signaling pathway in AGS cells.

I did not understand the sentence: CagA was not expressed in the AGS cells without H. pylori infection. CagA expression increased in the cells with H. pylori infection, but this expression dose-dependently decreased in the infected cells after eupatilin treatment. - did the authors modify the AGS lines?

6.Please increase the resolution of the figures. Currently they are unreadable

7.The request is to clearly describe the number of figures at the end of the manuscript.

8.The research methods are well selected.

Reviewer #2: The authors investigate the anti-inflammatory effects of eupatilin in gastric epithelial cells infected with Helicobacter pylori (H. pylori). The results show that eupatilin inhibits the translocation of CagA and suppresses the PI3K/NF-κB signaling pathway, thereby reducing the expression of pro-inflammatory cytokines in a dose-dependent manner. This anti-inflammatory effect suggests that eupatilin may play a protective role against H. pylori-induced gastritis, highlighting its potential as a therapeutic alternative in the context of increasing antibiotic resistance and possible gastric dysbiosis post-treatment. However, the study presents certain limitations that need to be addressed to improve the work.

Major:

1. To confirm the anti-inflammatory effects of eupatilin, the authors should conduct blocking or inhibition studies using siRNAs or specific blockers (e.g., PI3K/NF-κB) to corroborate if it indeed inhibits these molecular pathways.

2. Related to the previous point, another option would be to compare the results with those obtained using already established and safe therapies.

3. Despite strong in vitro evidence, the exact mechanisms by which eupatilin acts in vivo remain unclear. The interaction of H. pylori with the host involves complex host-environment factors, which are not fully replicated in in vitro studies. This point should be included as a limitation if the authors do not have in vivo study results.

4. While the study demonstrates the anti-inflammatory properties of eupatilin, its direct anti-tumor effects were not investigated. Future research should explore whether eupatilin has potential benefits in preventing or treating gastric cancer.

Minor:

1. In the different figures, HP should be defined in the figure legends.

2. Review Figure 4: Figures 4A and 4B are western blots, but the text describes something else.

3. Sometimes the text references Figs 4C–G and other times Fig 5A and 5B. Ensure all references have the same style.

Reviewer #3: Overview

This study is well structured and offers a deep insight into the anti-inflammatory potential of eupatilin on H. Pylori induced gastritis, a common gastric problem affecting a wide range of world population. I recommend the publication of this manuscript after addressing the following comments.

Comments

1. Please elaborate more on why a gastric carcinoma cell line was used in this study instead of a normal gastric cell line given that it already shows a heightened level of inflammation before incubation with CagA. If it is not possible to perform a control experiment using a normal cell line, please elaborate more on this issue in the discussion section by providing enough justification and references to the below part.

“Nevertheless, H. pylori- infected AGS cells can serve as a model for normal gastric epithelial cells exposed to H. pylori, given that AGS cells have been extensively employed in experiments on the response to H. pylori.”

2. The reference list is quite short for such a well-established disease model. Specifically, the materials and methods section does not include any references. Please add appropriate references for each of the methodologies used and for the primer sequences used in the PCR experiment.

3. Please add a reference for the hummingbird morphological changes expected after translocation as it seems like a crucial verification point of translocation in this model.

4. Figure 1 is missing the statistical p values on the bar graph.

5. Western blot figures are missing the ladder, specifically the exact weight of the isolated bands for all the assessed markers.

6. PLOS authors have the option to publish the peer review history of their article (what does this mean?). If published, this will include your full peer review and any attached files.

Reviewer #1: No

Reviewer #2: No

Reviewer #3: No

---

## [Author Response · Author response to Decision Letter 0]

26 Jul 2024

We truly appreciate your kind review and comments. We tried our best to revise our manuscript according to your recommendations and highlighted in red color for your convenience of re-reviewing. Details of our response to your comments are described as below. We hope our manuscript has been improved accordingly and is now suitable for publication in your journal. 

Major:

1. To confirm the anti-inflammatory effects of eupatilin, the authors should conduct blocking or inhibition studies using siRNAs or specific blockers (e.g., PI3K/NF-κB) to corroborate if it indeed inhibits these molecular pathways.

2. Related to the previous point, another option would be to compare the results with those obtained using already established and safe therapies.

-> Answer to question 1 and 2 : 

We truly appreciate your kind review and advices. First of all, we are very sorry for that we have difficulties in proceeding additional research including blocking/inhibition or comparison studies because of our current laboratory circumstances. Although your comments are really meaningful and we totally agree with your suggestions, there are practical issues since everything has to be redone from the beginning. 

However, we believe our research has supporting evidence of anti-inflammatory effects of eupatilin by suppressing the expression of PI3K/NF-κB for the following reason. We initially focused the inhibitory effect of eupatilin on H. pylori cagA (initial step of anti-inflammatory process) rather than the direct activity on PI3K/ NF-κB pathways. And it’s a very well-known hypothesis that H. pylori CagA activates PI3K/NF-κB signaling pathway. In our research, we determined that eupatilin treatment dose-dependently inhibited CagA expression, so we could demonstrate that eupatilin suppressed CagA-induced PI3K/ NF-κB pathways. 

We additionally described this issue in limitation as follows. 

Secondly, we did not confirm the direct effect of eupatilin on PI3K/NF-κB signaling. However, in this research, we initially focused the inhibitory effect of eupatilin on H. pylori cagA, a very well-known factor to activate PI3K/NF-κB signaling pathway, rather than the direct activity on PI3K/ NF-κB pathways. So we determined that eupatilin treatment dose-dependently inhibited CagA expression, which could sufficiently suppress CagA-induced PI3K/ NF-κB pathways.

We hope our explanation is sufficiently understandable and we look forward to hearing a positive response. Thank you so much for your kind comments and we will plan next research based on your advices. 

3. Despite strong in vitro evidence, the exact mechanisms by which eupatilin acts in vivo remain unclear. The interaction of H. pylori with the host involves complex host-environment factors, which are not fully replicated in in vitro studies. This point should be included as a limitation if the authors do not have in vivo study results.

-> Thank you for your kind review. We emphasized this limitation by adding your words in the revised manuscript. The revised part is as follows. 

Despite convincing in vitro evidence of the anti-inflammatory effects of eupatilin on H. pylori CagA-infected gastric epithelial cells, the exact mechanisms of action of eupatilin in vivo remains unclear. Since the interaction of H. pylori with the host involves complex host-environment factors, which could not be fully replicated in in vitro studies, further in vivo studies are needed to determine the safe and effective dose of eupatilin to exert anti-inflammatory effects against H. pylori-induced gastritis.

4. While the study demonstrates the anti-inflammatory properties of eupatilin, its direct anti-tumor effects were not investigated. Future research should explore whether eupatilin has potential benefits in preventing or treating gastric cancer.

-> Thank you for your kind review. We also emphasized the future research for direct anti-tumor effect of eupatilin by adding your words in the revised manuscript. The revised part is as follows. 

Future research should be conducted to reveal other anti-inflammatory pathways and identify the direct anti-tumor property whether eupatilin has potential effects on preventing or treating gastric cancer.

Minor:

1. In the different figures, HP should be defined in the figure legends.

-> Thank you for your detailed comment. We added the definition of HP (HP = Helicobacter pylori) in each figure legends.

2. Review Figure 4: Figures 4A and 4B are western blots, but the text describes something else.

-> We are sorry but we described the methods appropriately. Figure 4A is the results of western blot (black bands), and figure 4B is deduced by RT-PCR (white bands).

3. Sometimes the text references Figs 4C–G and other times Fig 5A and 5B. Ensure all references have the same style.

-> Thank you for your detailed comment. We corrected ‘4C-G’ into ‘4C, D, E, F, and G’.

---

## [Decision Letter · Decision Letter 1]

26 Aug 2024

PONE-D-24-23616R1Anti-inflammatory effects of eupatilin on Helicobacter pylori CagA-induced gastric inflammationPLOS ONE

Dear Dr. Kim,

Thank you for submitting your manuscript to PLOS ONE. After careful consideration, we feel that it has merit but does not fully meet PLOS ONE’s publication criteria as it currently stands. Therefore, we invite you to submit a revised version of the manuscript that addresses the points raised during the review process.

We look forward to receiving your revised manuscript.

Kind regards,

Saeid Ghavami, PhD

Academic Editor

PLOS ONE

Additional Editor Comments:

Dear Professor Kim,

Thank you for submitting the revised version of your manuscript. Upon reviewing the revision, I noticed that the responses to Reviewer 3's comments were not included. It is important to address these comments to ensure the manuscript is comprehensive and meets the reviewers' expectations.

Here are Reviewer 3's comments that need to be addressed:

Overview:

This study is well-structured and offers deep insight into the anti-inflammatory potential of eupatilin on H. pylori-induced gastritis, a common gastric problem affecting a wide range of the global population. I recommend the publication of this manuscript after addressing the following comments.

Comments:

Please elaborate more on why a gastric carcinoma cell line was used in this study instead of a normal gastric cell line, given that it already shows a heightened level of inflammation before incubation with CagA. If it is not possible to perform a control experiment using a normal cell line, please elaborate more on this issue in the discussion section by providing enough justification and references to the below part:

“Nevertheless, H. pylori-infected AGS cells can serve as a model for normal gastric epithelial cells exposed to H. pylori, given that AGS cells have been extensively employed in experiments on the response to H. pylori.”

The reference list is quite short for such a well-established disease model. Specifically, the materials and methods section does not include any references. Please add appropriate references for each of the methodologies used and for the primer sequences used in the PCR experiment.

Please add a reference for the hummingbird morphological changes expected after translocation, as it seems like a crucial verification point of translocation in this model.

Figure 1 is missing the statistical p-values on the bar graph.

Western blot figures are missing the ladder, specifically the exact weight of the isolated bands for all the assessed markers.

Kindly address these comments in your revision and resubmit the manuscript.

Please let me know if you need any further clarification. I look forward to receiving the revised manuscript.

Thank you for your attention to this matter.

Best regards,

Saeid Ghavami, PhD

Scientific Editor

Reviewers' comments:

Reviewer's Responses to Questions

**Comments to the Author**

1. If the authors have adequately addressed your comments raised in a previous round of review and you feel that this manuscript is now acceptable for publication, you may indicate that here to bypass the “Comments to the Author” section, enter your conflict of interest statement in the “Confidential to Editor” section, and submit your "Accept" recommendation.

Reviewer #1: All comments have been addressed

Reviewer #2: All comments have been addressed

2. Is the manuscript technically sound, and do the data support the conclusions?

Reviewer #1: Yes

Reviewer #2: Partly

3. Has the statistical analysis been performed appropriately and rigorously? 

Reviewer #1: Yes

Reviewer #2: Yes

4. Have the authors made all data underlying the findings in their manuscript fully available?

Reviewer #1: Yes

Reviewer #2: Yes

5. Is the manuscript presented in an intelligible fashion and written in standard English?

Reviewer #1: Yes

Reviewer #2: Yes

6. Review Comments to the Author

Reviewer #1: (No Response)

Reviewer #2: The authors have responded correctly and in detail to the comments. They have expressed their gratitude for the suggestions and explained why they cannot conduct additional studies due to their current laboratory circumstances. However, they have provided a justification based on their current research, arguing that the inhibition of CagA expression by eupatilin suggests that it might also suppress the CagA-induced PI3K/NF-κB pathways.

7. PLOS authors have the option to publish the peer review history of their article (what does this mean?). If published, this will include your full peer review and any attached files.

Reviewer #1: No

Reviewer #2: No

---

## [Author Response · Author response to Decision Letter 1]

9 Sep 2024

We truly appreciate your kind review and comments. We tried our best to revise our manuscript according to your recommendations and highlighted the revised parts in red for your convenience of re-reviewing. Details of our response to your comments are described below. We hope our manuscript has been improved accordingly and is now suitable for publication in your journal. 

1. Please elaborate more on why a gastric carcinoma cell line was used in this study instead of a normal gastric cell line, given that it already shows a heightened level of inflammation before incubation with CagA. If it is not possible to perform a control experiment using a normal cell line, please elaborate more on this issue in the discussion section by providing enough justification and references to the below part:

“Nevertheless, H. pylori-infected AGS cells can serve as a model for normal gastric epithelial cells exposed to H. pylori, given that AGS cells have been extensively employed in experiments on the response to H. pylori.”

-> Thank you for your comment. We additionally described this issue as a limitation in the discussion section as follows: 

We employed AGS human gastric carcinoma cell line instead of using normal gastric cell line, though AGS cells already could possess a heightened level of inflammation before H. pylori infection. Nevertheless, AGS cells have strong cell viability and have been extensively employed to understand the consequences of infection on epithelial cells, particularly in experiments on the response to H. pylori [16]. Therefore, we thought that AGS cells can serve as an alternative model for normal gastric epithelial cell in this research. 

(Reference 16. Matsunaga S, Nishiumi S, Tagawa R, Yoshida M. Alterations in metabolic pathways in gastric epithelial cells infected with Helicobacter pylori. Microb Pathog 2018;124:122-129.)

2. The reference list is quite short for such a well-established disease model. Specifically, the materials and methods section does not include any references. Please add appropriate references for each of the methodologies used and for the primer sequences used in the PCR experiment.

-> Thank you for your comment. We added references for methodologies used in the PCR experiment in part of ‘Analysis of mRNA expression’. (References 13,14,15)

3. Please add a reference for the hummingbird morphological changes expected after translocation, as it seems like a crucial verification point of translocation in this model.

-> Thank you for your comment. We added references for the hummingbird morphological changes expected after CagA translocation in part of ‘Effects of eupatilin on CagA translocation into AGS cells and CagA-induced morphologic changes’ (References 16,17)

4. Figure 1 is missing the statistical p-values on the bar graph.

-> Thank you for your comment. Since the purpose of cell viability test is not to compare the statistical value but to determine the treatment concentration (more than 80% of cell viability), we thought that it was not necessary to obtain p values. We referred to the research below to test for cell viability. 

Yeon MJ, Lee MH, Kim DH, et al. Anti-inflammatory effects of Kaempferol on Helicobacter pylori-induced inflammation. 2019;83166-173.

Choi EJ, Lee S, Char JR, et al. Eupatilin inhibits lipopolysaccharide-induced expression of inflammatory mediators in macrophages. Life Sci. 2011;88:1121-1126.

5. Western blot figures are missing the ladder, specifically the exact weight of the isolated bands for all the assessed markers.

-> Thank you for your detailed comment. We loaded samples onto one gel and cut the membrane for each molecular weight. We are sorry for not showing the ladder, but we indicated markers with the molecular weight in PPT file ‘Revision (marker)’. We can say that our experiments are reliable, and we hope our explanation is sufficiently understandable. We look forward to hearing a positive response.

---

## [Editor Report · Decision Letter 2]

4 Oct 2024

PONE-D-24-23616R2Anti-inflammatory effects of eupatilin on Helicobacter pylori CagA-induced gastric inflammationPLOS ONE

Dear Dr. Kim,

Thank you for submitting your manuscript to PLOS ONE. After careful consideration, we feel that it has merit but does not fully meet PLOS ONE’s publication criteria as it currently stands. Therefore, we invite you to submit a revised version of the manuscript that addresses the points raised during the review process.

Dear Professor Kim,

Thank you for your response. While I understand your rationale for not including p-values, I would strongly recommend reconsidering this approach. Even if the primary goal is to identify treatment concentrations that maintain cell viability above 80%, statistical analysis is still essential to ensure that the observed effects are significant and not due to random variation. Including p-values will add robustness to your conclusions and allow for a more scientifically rigorous interpretation of the results.

Statistical significance can help validate that the differences you observe between control and treated groups are meaningful, and it provides a clearer understanding of the treatment's impact. This is a standard expectation in most scientific publications, and adding this analysis will undoubtedly strengthen your manuscript.

I urge you to include appropriate statistical tests for your cell viability data to meet the required standards and ensure your results are well-supported.

Thank you for your attention to this matter.

Best regards,

Saeid Ghavami, PhD, AE

We look forward to receiving your revised manuscript.

Kind regards,

Saeid Ghavami, PhD

Academic Editor

PLOS ONE

Journal Requirements:

Additional Editor Comments :

Dear Professor Kim,

Thank you for your response. While I understand your rationale for not including p-values, I would strongly recommend reconsidering this approach. Even if the primary goal is to identify treatment concentrations that maintain cell viability above 80%, statistical analysis is still essential to ensure that the observed effects are significant and not due to random variation. Including p-values will add robustness to your conclusions and allow for a more scientifically rigorous interpretation of the results.

Statistical significance can help validate that the differences you observe between control and treated groups are meaningful, and it provides a clearer understanding of the treatment's impact. This is a standard expectation in most scientific publications, and adding this analysis will undoubtedly strengthen your manuscript.

I urge you to include appropriate statistical tests for your cell viability data to meet the required standards and ensure your results are well-supported.

Thank you for your attention to this matter.

Best regards,

Saeid Ghavami, PhD, AE
---

## [Author Response · Author response to Decision Letter 2]

18 Oct 2024

We appreciate your kind comment. We hope our manuscript has been improved accordingly and is now suitable for publication in your journal. 

1. Thank you for your response. While I understand your rationale for not including p-values, I would strongly recommend reconsidering this approach. Even if the primary goal is to identify treatment concentrations that maintain cell viability above 80%, statistical analysis is still essential to ensure that the observed effects are significant and not due to random variation. Including p-values will add robustness to your conclusions and allow for a more scientifically rigorous interpretation of the results.

Statistical significance can help validate that the differences you observe between control and treated groups are meaningful, and it provides a clearer understanding of the treatment's impact. This is a standard expectation in most scientific publications, and adding this analysis will undoubtedly strengthen your manuscript.

I urge you to include appropriate statistical tests for your cell viability data to meet the required standards and ensure your results are well-supported.

Thank you for your attention to this matter.

-> Thank you for your comment, and we added p-value in the revised figure (showing cell viability) according to your recommendation.

---

## [Editor Report · Decision Letter 3]

22 Oct 2024

Anti-inflammatory effects of eupatilin on Helicobacter pylori CagA-induced gastric inflammation

PONE-D-24-23616R3

Dear Dr. Kim,

We’re pleased to inform you that your manuscript has been judged scientifically suitable for publication and will be formally accepted for publication once it meets all outstanding technical requirements.

Kind regards,

Saeid Ghavami, PhD

Academic Editor

PLOS ONE
---

## [Editor Report · Acceptance letter]

25 Oct 2024

PONE-D-24-23616R3 

PLOS ONE

Dear Dr. Kim, 

I'm pleased to inform you that your manuscript has been deemed suitable for publication in PLOS ONE. Congratulations! Your manuscript is now being handed over to our production team.

Kind regards, 

on behalf of

Dr. Saeid Ghavami 

Academic Editor

PLOS ONE